# TimeFlow: Towards Stochastic-Aware and Efficient Time Series Generation via Flow Matching Modeling

## Abstract

Generating high-quality time series data has emerged as a critical research topic due to its broad utility in supporting downstream time series mining tasks. A major challenge lies in modeling the intrinsic stochasticity of temporal dynamics, as real-world sequences often exhibit random fluctuations and localized variations. While diffusion models have achieved remarkable success, their generation process is computationally inefficient, often requiring hundreds to thousands of expensive function evaluations per sample. Flow matching has emerged as a more efficient paradigm, yet its conventional ordinary differential equation (ODE)-based formulation fails to explicitly capture stochasticity, thereby limiting the fidelity of generated sequences. By contrast, stochastic differential equation (SDE) are naturally suited for modeling randomness and uncertainty. Motivated by these insights, we propose **TimeFlow**, a novel SDE-based flow matching framework that integrates a encoder-only architecture. Specifically, we design a component-wise decomposed velocity field to capture the multifaceted structure of time series and augment the vanilla flow-matching optimization with an additional stochastic term to enhance representational expressiveness. TimeFlow is flexible and general, supporting both unconditional and conditional generation tasks within a unified framework. Extensive experiments across diverse datasets demonstrate that our model consistently outperforms strong baselines in generation quality, diversity, and efficiency. The code is available at https://anonymous.4open.science/r/TimeFlow-59E4.

## 1 Introduction

Time series generation plays an important role across many real-world domains, including finance, healthcare, and energy management (Lim & Zohren, 2021; Vuletić et al., 2024; Deng et al., 2025; Lin et al., 2025), where reliable modeling and simulation are essential for decision making, intelligent management, and risk assessment (Alaa et al., 2021; Cheng et al., 2025). Consequently, time series generation has attracted considerable research attention. Unlike images or text, time series are characterized by long-range dependencies (Ubal et al., 2023) and inherent stochasticity (Luo et al., 2025; Wang et al., 2025), which make their generative modeling particularly challenging. A central challenge in this field lies in capturing the inherent stochasticity of temporal dynamics, which arises from noise and perturbations and is crucial for reproducing realistic variability (Cheng et al., 2023).

A variety of generative frameworks have been explored for time series modeling, including generative adversarial networks (GANs) (Goodfellow et al., 2014) and variational autoencoders (VAEs) (Kingma & Welling, 2013), which provide early probabilistic and adversarial approaches for synthesizing temporal data. More recently, denoising diffusion probabilistic models (DDPMs) (Song et al., 2021; Ho et al., 2020) have achieved remarkable success in the broader field of generative modeling, powering breakthroughs in image, speech, and scientific data synthesis (Jiang et al., 2025; Tai et al., 2023; Yuan & Qiao, 2024). They have also demonstrated strong potential for time series tasks, setting new baselines for generative quality. Despite their impressive flexibility, DDPMs are computationally inefficient, as iterative denoising requires hundreds to thousands of reverse steps (Figure 1(a)). This inefficiency is particularly problematic in time series generation, where longer sequences further amplify the computational burden, limiting the practical-

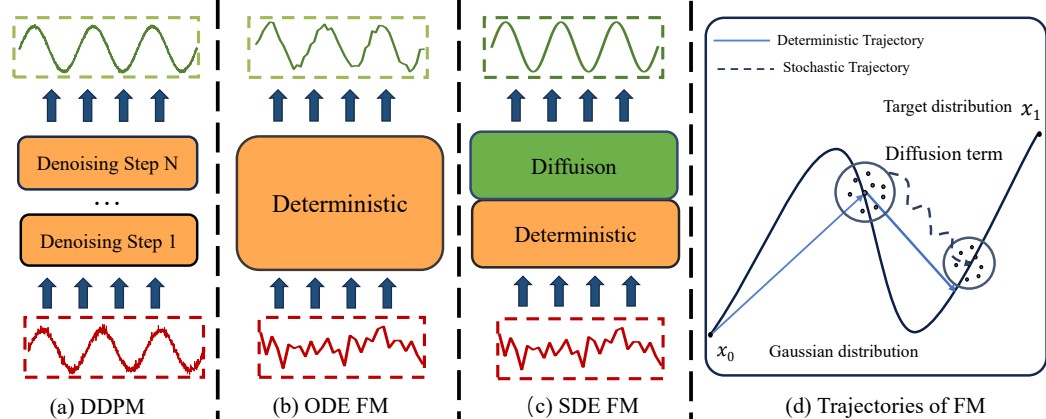

Figure 1: Comparison of generative paradigms. (a) DDPM rely on iterative denoising. (b) ODE-based flow matching produces deterministic trajectories. (c) SDE-based flow matching yields stochastic trajectories that capture uncertainty via diffusion terms. (d) Illustration of deterministic versus stochastic trajectories under flow matching.

ity of DDPMs in large-scale or latency-sensitive scenarios. To overcome this efficiency bottleneck, flow matching (FM) (Lipman et al., 2022) has emerged as a promising alternative by directly parameterizing continuous generative trajectories through velocity fields, offering both stable training objectives and efficient sampling mechanisms.

However, most existing FM applications adopt an ordinary differential equation (ODE) formulation, where trajectories are determined by deterministic velocity fields, as illustrated in Figure 1(b). While effective in modeling global structure, this ODE-based FM model suffers from two key limitations for time series generation. First, it does not explicitly model stochasticity, resulting in synthetic sequences that lack variability and fail to reproduce rare but important fluctuations. Second, by constraining the generative dynamics to a deterministic flow, the model cannot adequately represent predictive uncertainty or adapt to heterogeneous temporal regimes. In other words, although FM improves efficiency, the ODE formulation sacrifices the ability to faithfully reflect the stochastic nature of real-world temporal processes. These shortcomings motivate extending FM to a stochastic differential equation (SDE) formulation, where diffusion terms (Figure 1(c)) enable richer variability, uncertainty-aware trajectories, and improved robustness to random perturbations.

To address these challenges, we propose TimeFlow, an SDE-based flow matching framework for efficient and stochasticity aware time series generation (Figure 1(d)). By incorporating a diffusion term into the generative process, TimeFlow explicitly captures randomness and produces higher-quality trajectories with faithful uncertainty modeling. To further preserve temporal structures, we employ a transformer-based encoder with decomposition mechanism, which maintains consistency with underlying dynamics while keeping the framework and efficient. Moreover, TimeFlow naturally extends to both unconditional and conditional generation tasks, including forecasting and imputation, thereby broadening its applicability in practice.

In summary, our major contributions are as follows:

- We propose TimeFlow, a novel time series generation framework that extends flow matching to a stochastic differential equation formulation. By introducing the Stochastic Flow Matching loss, our method explicitly models temporal stochasticity, enabling uncertainty aware generative processes that better capture complex random fluctuations.

- We leverage the flow matching paradigm to fundamentally alleviate the efficiency bottlenecks of diffusion-based approaches, enabling significantly faster generation of high dimensional time series while maintaining both reliability and fidelity.

- We conduct evaluations on both unconditional and conditional generation tasks, including forecasting and imputation, across diverse real world datasets. Experimental results demonstrate that TimeFlow outperforms strong baselines in both quality and efficiency.

## 2 RELATED WORK

**Time Series Generation.** Generative models have recently shown strong performance in time series modeling. Early efforts focused on GAN-based methods, such as TimeGAN (Yoon et al., 2019), which introduces a supervised embedding to preserve temporal dynamics. COT-GAN (Xu et al., 2020) improves the stability of training through causal optimal transport and entropic regularization, while GT-GAN (Jeon et al., 2022) offers a general framework for regular and irregular time series. VAE-based approaches were later explored. TimeVAE (Desai et al., 2021) proposes an interpretable structure tailored to time series, and CR-VAE (Li et al., 2023) encodes causal dependencies via a sparse critical matrix. More recently, diffusion models have emerged as a leading paradigm. DiffTime (Coletta et al., 2023) decouples constraint specification from training, enabling flexible inference-time adaptation. SSSD (Alcaraz & Strodthoff, 2022) and CSDI (Tashiro et al., 2021) extend the framework to conditional tasks using self-supervised masking. TimeGrad (Rasul et al., 2021) applies RNN-guided autoregression, while TSDiff (Shen & Kwok, 2023) enables unconditional generation via self-guidance. Finally, Diffusion-TS (Yuan & Qiao, 2024) introduces an interpretable design that disentangles temporal semantics.

**Flow Matching.** As a promising alternative to diffusion processes, flow matching (Lipman et al., 2022) has shown significant potential in generative modeling due to its training stability, flexible trajectory design, and computational efficiency. Flow matching was initially proposed for image generation tasks, where it effectively modeled complex data distributions (Dao et al., 2023; Stoica et al., 2025; Yazdani et al., 2025; Geng et al., 2025). It and its variants, such as rectified flow (Liu, 2022) and OT flow matching (Klein et al., 2023)—have since been extended beyond image generation to a variety of domains, including video (Jin et al., 2024; Cao et al., 2025; Davtyan et al., 2023), text (Hu et al., 2024), and audio (Jung et al., 2024). Recently, flow matching has been applied to time series. TFM (Zhang et al., 2024) uses neural SDEs to tackle stochastic and irregularly sampled clinical time series forecasting. SGFM (He et al., 2024) combines state space models and GNNs for refined anomaly detection. CGFM (Xu et al., 2025) models prediction time series forcasting errors by conditional guidance. CFM-TS (Tamir et al., 2024) applies conditional probability paths to neural ODEs for time series modeling.

## 3 PRELIMINARIES

### 3.1 PROBLEM STATEMENT

Let $\mathbf{X}_{1:t} = (x_1, \ldots, x_t) \in \mathbb{R}^{\tau \times d}$ denote an observed time series, where $t$ represents the number of time steps, and $d$ denotes the dimension of each observation. Given the time series dataset $\mathcal{S} = \{\mathbf{X}_{1:t}^i\}_{i=1}^N$ consisting of $N$ time series samples, the objective of the unconditional generation task is to learn a flow-based generator $G$ to synthesize sequences $\hat{\mathbf{X}}_{1:t}^i = G(\mathbf{S}_i)$, which closely approximates the realistic time series data $\mathbf{X}_{1:t}^i$. And the goal of conditional generation is to generate samples from a conditional distribution $p(\cdot \mid y)$, where $y$ is a control variable that can be any real-world signal and dictates the synthesis.

### 3.2 FLOW MATCHING

Flow matching (Lipman et al., 2022) is a family of generative models that learn to match probability flows represented by velocity fields between a simple prior distribution and the data distribution. Formally, given data $x_1 \sim p_{\text{data}}(x)$ and a prior noise variable $x_0 \sim p_{\text{prior}}(\epsilon)$, a flow path can be defined as $x_t = a_t x_1 + b_t x_0$, where $a_t$ and $b_t$ are predefined schedules. The corresponding velocity is given by $v_t = \dot{z}_t = a_t' x + b_t' \epsilon$, with $(\cdot)'$ denoting the derivative with respect to time. This is referred to as the conditional velocity, written as $v_t = v_t(x_t \mid x_1)$. A common choice of schedules is $a_t = t$ and $b_t = 1 - t$, which yields $v_t = x_1 - x_0$.

Since a given $x_t$ and its associated velocity $v_t$ can arise from different pairs of $x$ and $\epsilon$, flow mtching essentially models the expectation over all possibilities, which is called the marginal velocity: $\nu(x_t, t) = \mathbb{E}_{p(v_t|x_t)}[v_t]$. A neural network $v_\theta$ parameterized by $\theta$ is trained to approximate this marginal velocity field. The conditional flow matching loss is written as:

$$\mathcal{L}_{\text{CFM}}(\theta) = \mathbb{E}_{t,x_1,x_0}\left[\|v_\theta(x_t, t) - v_t\|^2\right], \tag{1}$$

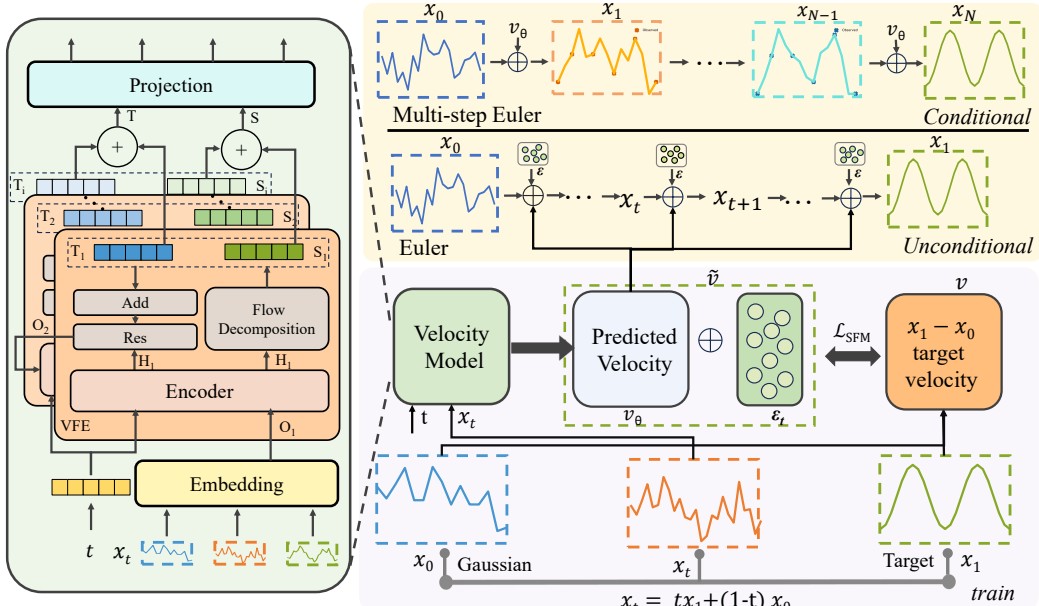

Figure 2: Overview of the proposed TimeFlow architecture

where the target $v_t$ corresponds to the conditional velocity. By minimizing this objective, the learned velocity field aligns with the expected flow, enabling efficient training of generative models.

### 3.3 NEURAL STOCHASTIC DIFFERENTIAL EQUATION

Neural stochastic differential equations describe the evolution of a latent state with both deterministic and stochastic components, given by:

$$dz_t = f_\theta(x_t, t)\, dt + g_\theta(x_t, t)\, dW_t, \tag{2}$$

where $f_\theta$ denotes the drift term parameterized by a neural network, $g_\theta$ is the diffusion coefficient, and $W_t$ is a standard Wiener process. The first term $f_\theta(x_t, t)\, dt$ corresponds to the deterministic drift, while the second term $g_\theta(x_t, t)\, dW_t$ represents stochastic perturbations from Brownian motion.

## 4 METHODS

### 4.1 OVERVIEW

Time series data often exhibit inherent stochasticity caused by noise and random perturbations. To address this issue, we propose a flow matching framework that explicitly learns velocity fields to align the dynamics between prior noise and target sequences. As illustrated in Figure 2, our method consists of four main components. First, the velocity model, composed of multiple velocity field encoder (VFE) layers, parameterizes the continuous transformation from a Gaussian prior $x_0$ to the real data distribution $x_1$. Second, stochastic optimization objective incorporates perturbations $\epsilon_t$ to capture the inherent randomness of time series. Third, flow decomposition (FD) employs decomposition to disentangle temporal components. Finally, stochastic sampling enables versatile generation.Unconditional generation directly synthesizes sequences from noise, while conditional generation exploits observed data to produce consistent and plausible completions.

### 4.2 VELOCITY MODEL

We propose a velocity model consisting of multiple Velocity Field Encoder (VFE) layers, each instantiated as a transformer encoder architecture that captures long dependency and global contextual interactions. At the input stage, a convolutional embedding layer extracts local temporal patterns and

projects raw sequences into a high-dimensional representation space. Then, the encoder layer refines temporal representations and applies a decomposition mechanism to disentangle the output into two complementary components. The seasonal component captures high-frequency stochastic dynamics, while the trend component represents low-frequency smooth evolution as a stable baseline. This explicit separation prevents heterogeneous sources of uncertainty from being conflated into a single representation, thereby enabling a more structured parameterization of the velocity field and enhancing robustness to stochastic perturbations.

**Flow Decomposition.** In our velocity field encoder, the encoder output $H_i$ is first decomposed via moving average into a trend component $t_i$ and a seasonal component $s_i$ for subsequent processing:

$$t_i = \text{AvgPool}(\text{Padding}(H_i)), \qquad s_i = H_i - t_i, \tag{3}$$

Given that seasonal patterns are vulnerable to contextual disturbances such as phase shifts and amplitude variations, we refine $s_i$ using a cross-attention (CA) mechanism with the original representation $H_i$, and apply a residual connection to promote stable learning. In comparison, the trend component $t_i$ is inherently more stable; nevertheless, we further modulate it through a multi-scale gated convolutional module that effectively incorporates seasonal information:

$$S_i = \text{CA}(s_i, H_i) + s_i, \qquad T_i = \Big( \sum_{k=1}^{K} \text{Conv1D}_k(S_i) \Big) \odot t_i, \tag{4}$$

Here, $S_i$ represents the refined seasonal representation, and $T_i$ the modulated trend representation. Subsequently, the residual representation for the next layer is computed by removing both components from the encoder output, whereas the velocity field is generated from the aggregated seasonal–trend terms accumulated over $D$ layers of the encoder:

$$O_{i+1} = H_i - S_i - T_i, \qquad v_\theta(x_t, t) = \text{Projection}\Big( \sum_{i=1}^{D} S_i + \sum_{i=1}^{D} T_i \Big), \tag{5}$$

where $O_{i+1}$ is the residual representation passed to the $(i+1)$-th encoder layer, and $v_\theta(x_t, t)$ is the velocity field parameterized by $\theta$ at time step $t$.

## 4.3 STOCHASTIC OPTIMIZATION OBJECTIVE

To explicitly account for stochastic perturbations in time series, we introduce the stochastic flow matching (SFM) loss. As illustrated in Figure 2, given an intermediate state $x_t$, the velocity model predicts a velocity field $v_\theta(x_t, t)$. To model uncertainty, a Gaussian noise term $\epsilon_t \sim \mathcal{N}(0, \sigma_t^2 I)$ is injected into the prediction, yielding a perturbed velocity representation: $\tilde{v}(x_t, t) = v_\theta(x_t, t) + \epsilon_t$. The SFM loss is then defined as:

$$\mathcal{L}_{\text{SFM}} = \mathbb{E}_{t, x_t} \Big[ \| \tilde{v}(x_t, t) - v_t \|^2 \Big], \tag{6}$$

where $v_t = x_1 - x_0$ denotes the ground-truth velocity between the initial and terminal states. By minimizing $\mathcal{L}_{\text{SFM}}$, the model is encouraged to learn a velocity field that captures not only deterministic dynamics but also stochastic variations, thereby facilitating robust and faithful modeling of uncertain time series.

## 4.4 STOCHASTIC SAMPLING

As illustrated in Figure 2, we design two distinct sampling flows for time series generation: unconditional and conditional. Both flows rely on the learned velocity field but differ in how stochastic perturbations and observational information are incorporated, thereby providing complementary perspectives on how the model handles uncertainty. In particular, unconditional flow emphasizes the role of randomness in generating diverse trajectories, while conditional flow demonstrates how partial observations can guide the generation process toward consistency with known information.

**Unconditional Generation.** The unconditional generation process begins from a Gaussian prior $X_0 \sim \mathcal{N}(0, I)$ and evolves according to a stochastic integral equation of the form:

$$X_1 = \int_0^1 \Big( v_\theta(t, X_t) \, dt + \sigma \, dW_t \Big), \tag{7}$$

where $v_\theta(t, X_t)$ denotes the learned velocity field, and $\sigma \, dW_t$ is a stochastic term driven by Brownian motion that introduces random perturbations. This formulation explicitly establishes a connection between the generative process and stochastic dynamics, ensuring that the synthesized trajectories not only capture the overall structure of time series but also reproduce their intrinsic variability. By integrating both deterministic velocity guidance and stochastic diffusion.

**Conditional Generation.** The conditional generation process incorporates partial observations of the target sequence as guidance. At each iteration, the integration time $t \in [0, 1]$ is reparameterized using a power-based sampling scheme, which controls the pace of noise injection and enables both early stochastic exploration and stable refinement in later stages. The latent state is obtained by interpolating between Gaussian noise and the observed sequence, with entries specified by a partial mask replaced by noisy versions of the ground truth to enforce conditional fidelity. The evolution of the trajectory is then governed by an Euler update rule of the form:

$$x_{t+1} = x_t + (1 - t) \, v_\theta(x_t, t), \tag{8}$$

where $v_\theta(\cdot)$ denotes the learned velocity field. To ensure stability, the outputs are further bounded through clamping. This formulation generates trajectories that remain faithful to observed data while preserving realistic variability in unobserved regions.

## 5 EXPERIMENTS

### 5.1 EXPERIMENTAL SETUPS

**Dataset.** We evaluate our approach on four real-world datasets and two synthetic datasets. The real-world datasets include Stocks, which contains Google stock prices from 2004 to 2019; ETTh, which records 15-minute electricity transformer load and oil temperature measurements; and UCI Energy, which provides household appliance energy consumption data. In addition, the fMRI dataset offers simulated BOLD signals. For synthetic benchmarks, we adopt the Sines dataset, comprising five-dimensional sinusoidal sequences, and the MuJoCo dataset, which generates multivariate time series from a physics-based simulation environment.

**Metrics.** We evaluate the quality of the synthesized data using four widely adopted metrics. (1) The discriminative score (Yoon et al., 2019) measures the similarity between real and synthetic data based on a supervised classification task. (2) The predictive score (Yoon et al., 2019) assesses the utility of synthetic data by training a sequence model on synthetic samples and evaluating it on real data. (3) The Context-Fréchet Inception Distance (Context-FID) (Jeha et al., 2022) quantifies the sample quality by comparing their contextual representations with those of real data. (4) The correlational score (Jeha et al., 2022) evaluates temporal dependency by computing the absolute error between correlation structures of the real and synthetic data.

**Baselines.** For unconditional generation, we compare our model with six baselines, including Diffusion-TS (Yuan & Qiao, 2024), TimeGAN (Yoon et al., 2019), TimeVAE (Desai et al., 2021), DiffWave (Kong et al., 2020), DiffTime (Coletta et al., 2023), and Cot-GAN (Xu et al., 2020). For conditional generation, we compare our model with Diffusion-TS (Yuan & Qiao, 2024) and CSDI (Tashiro et al., 2021).

### 5.2 UNCONDITIONAL TIME SERIES GENERATION

In Table 1, we present results for 24-length time series generation, a setting widely adopted in prior studies. TimeFlow consistently outperforms baseline methods across six datasets and nearly all evaluation metrics, with average improvements of 59.9% in Context-FID and 57.5% in Discriminative Score over the strongest competitor. We further evaluate TimeFlow-ODE, an ODE-based variant, which achieves competitive performance and attains state-of-the-art results in more than half of the cases. Nonetheless, TimeFlow consistently surpasses TimeFlow-ODE. These findings show that explicitly modeling stochasticity improves robustness and fidelity, highlighting the necessity of capturing intrinsic randomness in time series.

Table 1: Results on multiple time-series datasets (Bold indicates best performance).

| Metric (↓) | Methods | Sincs | Stocks | ETTh | MuJoCo | Energy | fMRI |
|---|---|---|---|---|---|---|---|
| Context-FID Score Lower Better | TimeFlow | **0.002±0.000** | **0.011±0.004** | **0.016±0.001** | **0.012±0.001** | **0.027±0.003** | 0.122±0.009 |
| | TimeFlow-ODE | 0.007±0.001 | 0.016±0.004 | 0.082±0.015 | 0.037±0.005 | 0.034±0.003 | 0.170±0.008 |
| | Diffusion-TS | 0.006±0.000 | 0.147±0.025 | 0.116±0.010 | 0.013±0.001 | 0.089±0.024 | **0.105±0.006** |
| | TimeGAN | 0.101±0.014 | 0.103±0.013 | 0.300±0.013 | 0.563±0.052 | 0.767±0.103 | 1.292±0.218 |
| | TimeVAE | 0.307±0.060 | 0.215±0.035 | 0.805±1.186 | 0.251±0.015 | 1.631±1.142 | 14.449±9.969 |
| | Diffwave | 0.014±0.002 | 0.232±0.032 | 0.873±0.061 | 0.393±0.041 | 1.031±1.131 | 0.244±0.018 |
| | DiffTime | 0.006±0.001 | 0.236±0.074 | 0.299±0.044 | 0.188±0.028 | 0.279±0.045 | 0.340±0.015 |
| | Cot-GAN | 1.337±0.068 | 0.408±0.086 | 0.980±0.071 | 1.094±0.079 | 1.039±0.028 | 7.813±3.550 |
| Correlational Score Lower Better | TimeFlow | **0.011±0.001** | **0.003±0.002** | **0.027±0.010** | **0.163±0.032** | 0.576±0.103 | **0.837±0.010** |
| | TimeFlow-ODE | 0.017±0.007 | 0.008±0.005 | 0.048±0.014 | 0.170±0.022 | **0.573±0.094** | 0.930±0.016 |
| | Diffusion-TS | 0.015±0.004 | 0.004±0.001 | 0.049±0.008 | 0.193±0.027 | 0.856±1.147 | 1.411±0.42 |
| | TimeGAN | 0.045±0.010 | 0.063±0.005 | 0.210±0.006 | 0.886±0.039 | 4.010±1.104 | 23.506±2.039 |
| | TimeVAE | 0.131±0.010 | 0.095±0.008 | 0.111±0.200 | 0.388±0.041 | 1.688±2.226 | 17.292±3.526 |
| | Diffwave | 0.022±0.005 | 0.030±0.020 | 0.175±0.006 | 0.579±0.018 | 5.001±1.154 | 3.927±0.409 |
| | DiffTime | 0.017±0.004 | 0.008±0.002 | 0.067±0.005 | 0.128±0.031 | 1.158±0.095 | 1.501±0.048 |
| | Cot-GAN | 0.049±0.010 | 0.007±0.004 | 0.249±0.009 | 1.041±2.007 | 3.164±1.061 | 26.824±4.449 |
| Discriminative Score Lower Better | TimeFlow | **0.004±0.004** | **0.011±0.010** | **0.010±0.006** | **0.005±0.007** | 0.061±0.007 | **0.101±0.015** |
| | TimeFlow-ODE | 0.014±0.007 | 0.034±0.018 | 0.023±0.010 | 0.023±0.026 | **0.060±0.021** | 0.141±0.019 |
| | Diffusion-TS | 0.006±0.007 | 0.067±0.015 | 0.061±0.009 | 0.008±0.002 | 0.122±0.003 | 0.167±0.023 |
| | TimeGAN | 0.011±0.008 | 0.102±0.021 | 0.114±0.055 | 0.238±0.068 | 0.236±0.012 | 0.484±0.042 |
| | TimeVAE | 0.041±0.044 | 0.145±1.120 | 0.209±0.058 | 0.230±1.102 | 0.499±1.000 | 0.476±0.044 |
| | Diffwave | 0.017±0.008 | 0.232±0.061 | 0.190±0.008 | 0.203±0.096 | 0.493±0.004 | 0.402±0.029 |
| | DiffTime | 0.013±0.006 | 0.097±0.016 | 0.100±0.007 | 0.154±0.045 | 0.445±0.004 | 0.245±0.051 |
| | Cot-GAN | 0.254±1.137 | 0.230±0.016 | 0.325±0.099 | 0.426±0.022 | 0.498±0.002 | 0.492±0.018 |
| Predictive Score Lower Better | TimeFlow | **0.093±0.000** | 0.037±0.000 | **0.119±0.002** | 0.008±0.000 | 0.250±0.000 | **0.099±0.000** |
| | TimeFlow-ODE | 0.093±0.000 | 0.037±0.000 | 0.123±0.009 | 0.011±0.000 | 0.250±0.000 | 0.099±0.000 |
| | Diffusion-TS | 0.093±0.000 | **0.036±0.000** | 0.119±0.002 | **0.007±0.003** | 0.250±0.000 | 0.099±0.000 |
| | TimeGAN | 0.093±0.019 | 0.038±0.001 | 0.124±0.001 | 0.025±1.000 | 0.273±0.004 | 0.126±1.002 |
| | TimeVAE | 0.093±0.000 | 0.039±0.000 | 0.126±0.004 | 0.012±0.002 | 0.292±0.000 | 0.113±0.003 |
| | Diffwave | 0.093±0.000 | 0.047±0.001 | 0.130±0.001 | 0.013±0.001 | 0.251±0.000 | 0.101±0.000 |
| | DiffTime | 0.093±0.000 | 0.038±0.000 | 0.121±0.004 | 0.010±0.000 | 0.252±0.000 | 0.100±0.000 |
| | Cot-GAN | 0.100±0.000 | 0.047±0.001 | 0.129±0.000 | 0.068±0.009 | 0.259±0.000 | 0.185±0.003 |
| | Original | 0.094±.001 | 0.036±.001 | 0.121±.005 | 0.007±.001 | 0.250±.003 | 0.090±.001 |

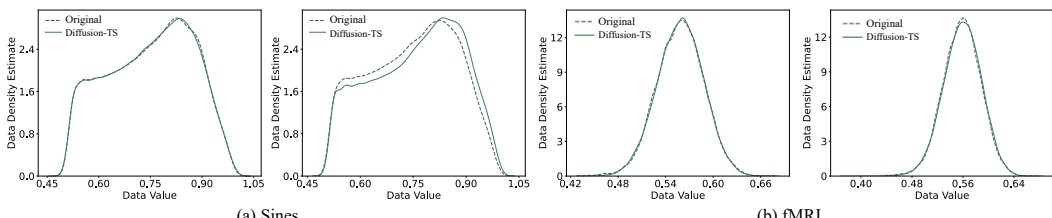

(a) Sines      (b) fMRI

Figure 3: KDE analysis of time series synthesized by TimeFlow and Diffusion-TS

To evaluate the quality of synthetic data, we conduct PCA, KDE, and t-SNE analyzes. As shown in Figure 4, PCA visualizations provide insights into the alignment between generated samples and real data in the reduced feature space. TimeFlow demonstrates a significantly closer clustering with the original distribution compared to Diffusion-TS, suggesting its stronger ability to retain the global variance structure of time series. The KDE plots in Figure 4 further confirm this finding, showing that the marginal distributions of sequences produced by TimeFlow closely match the ground truth. These results demonstrate that TimeFlow generates more realistic and distributionally aligned time series, effectively capturing both global and local dynamics. Additional t-SNE visualizations are provided in the Appendix D.

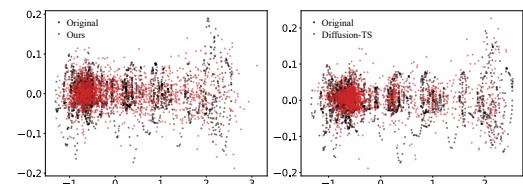

Figure 4: PCA analysis on Stocks dataset

## 5.3 CONDITION TIME SERIES GENERATION

We conduct extensive experiments to evaluate our model on both forecasting and imputation tasks. Figure 5 presents results on the Energy and Stocks datasets with sequence length 48. For the imputation task, as the missing ratio increases from 0.1 to 0.9, TimeFlow consistently achieves lower MSE than the baselines, indicating its robustness in handling stochastic perturbations under severe data sparsity. For the forecasting task, as the prediction window expands from 6 to 24, our model maintains lower MSE than Diffusion-TS and CSDI, demonstrating its effectiveness in capturing intrinsic randomness in temporal dynamics while preserving long range predictive accuracy.

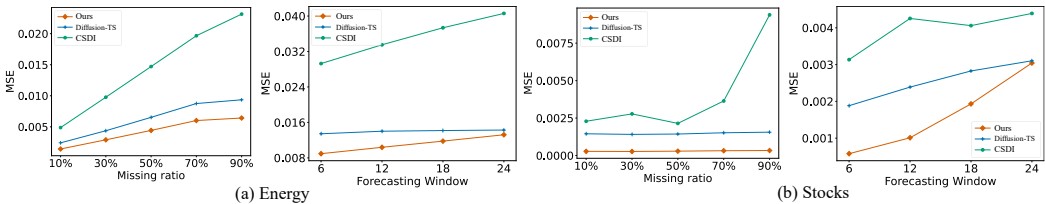

Figure 5: Performance of various methods for time-series imputation and forecasting.

To further assess the reliability of our method, we provide qualitative visualizations on the Energy and Mujoco datasets in Figure 6. The solid line denotes the median trajectory, while the shaded region indicates the 5%–95% quantile interval, capturing the uncertainty of the generated sequences. In the imputation task with a missing ratio of 0.9, our method successfully reconstructs coherent temporal patterns and smoothly recovers missing segments under extremely sparse observations. In the forecasting task with sequence length 48 and prediction length 12, our model produces trajectories more consistent with the ground truth and with fewer deviations than Diffusion-TS.

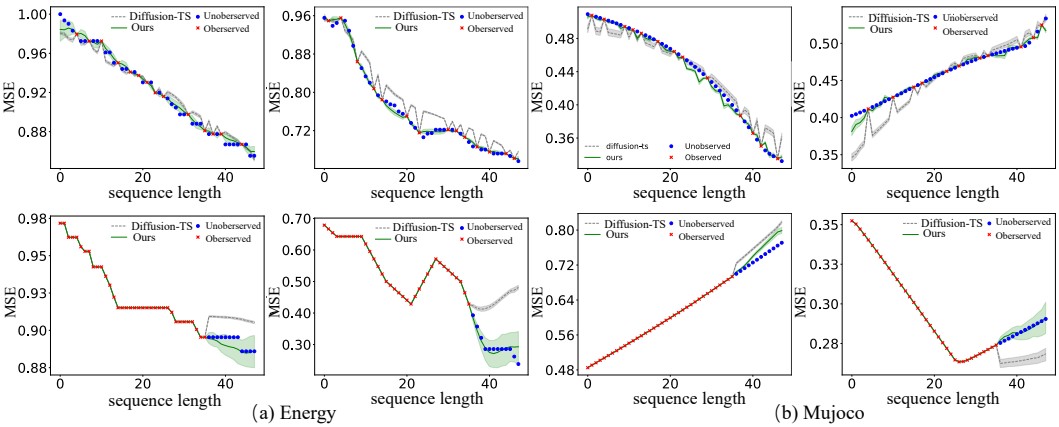

Figure 6: Examples of time series imputation and forcasting for Energy and Mujoco datasets.

## 5.4 EFFECT OF DIFFUSION COEFFICIENT

We further investigate the impact of the constant diffusion coefficient in conditional generation. As shown in Figure 7, results on the Energy and ETTh datasets reveal distinct behaviors. Energy is a high-dimensional dataset with stable dynamics, where the influence of diffusion remains marginal and performance is largely consistent across noise levels. In contrast, ETTh has lower dimensionality with stronger variability, making the choice of diffusion coefficient critical. Moderate noise yields more reliable results by balancing uncertainty modeling and trajectory stability, whereas very small or large noise leads to underfitting or instability. These findings highlight that the effectiveness of noise injection depends on dataset characteristics, underscoring the importance of adaptive uncertainty modeling for robust conditional generation.

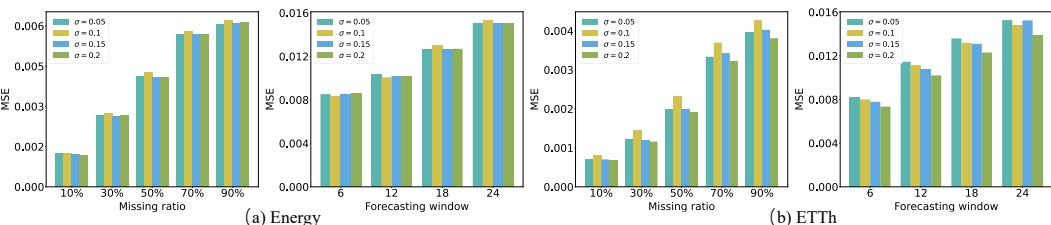

Figure 7: Impact of diffusion coefficient on conditional generation performance

## 5.5 ANALYSIS OF EFFICIENCY

To validate the efficiency of our model, we comprehensively compare TimeFlow with Diffusion-TS across different sampling steps and diverse datasets. On Stocks and Mujoco, TimeFlow requires significantly less sampling time and achieves superior Context-FID under the same step size (Figure 8). Moreover, across multiple datasets, TimeFlow consistently demonstrates clear efficiency advantages, with lower training costs and faster sampling than Diffusion-TS (Table 2).

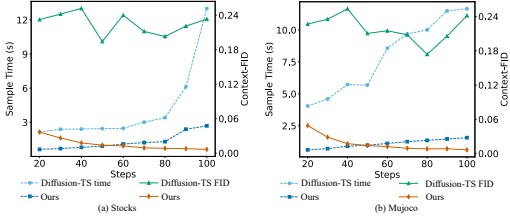

Figure 8: Analysis of efficiency and generation metrics on Stocks dataset

| Methods | Stage | Stocks | MuJoCo | Energy | fMRI |
|---|---|---|---|---|---|
| TimeFlow | train | 373.88 | 600.18 | 1112.6 | 453.27 |
| | uncondition | 2.69 | 5.16 | 25.52 | 5.69 |
| | forecasting | 2.54 | 4.86 | 9.14 | 1.56 |
| | imputation | 2.08 | 4.73 | 9.22 | 3.62 |
| Diffusion-ts | train | 687.69 | 873.39 | 1690.77 | 983.48 |
| | uncondition | 3.08 | 7.65 | 28.11 | 17.69 |
| | forecasting | 29.41 | 39 | 123.17 | 71.84 |
| | imputation | 29.99 | 41.36 | 101.91 | 122.02 |

Table 2: Training and sampling time comparison on different datasets in seconds (s).

## 5.6 ABLATION

To evaluate the effectiveness of the proposed TimeFlow model, we conduct an ablation study by removing three key components: (1) w/o CA: removing cross-attention in the flow decomposition; (2) w/o FD: removing the decomposition mechanism; and (3) w/o Encoder: removing the self-attention backbone. The results are summarized in Table 3. We observe that the complete TimeFlow consistently delivers superior or competitive performance across all datasets and evaluation metrics. Removing the CA or FD leads to only a moderate degradation in performance. Interestingly, on high-dimensional datasets such as fMRI, eliminating the encoder unexpectedly improves performance. We attribute this to the fact that, in extremely high-dimensional scenarios, the encoder may introduce excessive complexity and amplify noise, which can hinder generalization.

Table 3: Ablation study for model architecture and options. (Bold indicates best performance).

| Metric | Methods | Sines | Stocks | ETTh | MuJoCo | Energy | fMRI |
|---|---|---|---|---|---|---|---|
| Discriminative Score | TimeFlow | **0.004±0.004** | **0.011±0.001** | **0.010±0.006** | **0.005±0.007** | **0.061±0.007** | 0.101±0.015 |
| | w/o CA | 0.005±0.004 | 0.017±0.013 | 0.014±0.012 | 0.010±0.010 | 0.063±0.014 | 0.094±0.016 |
| | w/o FD | 0.005±0.005 | 0.092±0.029 | 0.013±0.005 | 0.012±0.019 | 0.062±0.013 | 0.149±0.030 |
| | w/o Encoder | 0.010±0.004 | 0.104±0.009 | 0.012±0.007 | 0.101±0.077 | 0.142±0.052 | **0.067±0.098** |
| Predictive Score | TimeFlow | **0.093±0.000** | **0.037±0.000** | **0.119±0.002** | **0.008±0.000** | **0.250±0.000** | **0.099±0.000** |
| | w/o CA | **0.093±0.000** | **0.037±0.000** | 0.121±0.004 | 0.010±0.000 | 0.251±0.000 | 0.100±0.000 |
| | w/o FD | **0.093±0.000** | 0.038±0.000 | 0.123±0.003 | 0.011±0.000 | 0.251±0.000 | 0.100±0.000 |
| | w/o Encoder | 0.093±0.001 | 0.039±0.000 | 0.121±0.005 | 0.015±0.002 | 0.252±0.000 | 0.100±0.000 |

## 6 CONLUSION

In this paper, we propose TimeFlow, a novel flow matching framework under the SDE paradigm for time series generation. By introducing a constant diffusion coefficient, TimeFlow effectively captures stochasticity and improves robustness. The flow matching formulation enables efficient training and sampling while maintaining high-quality generation. Experiments on multiple datasets demonstrate its effectiveness in both unconditional and conditional settings. Future work will explore more efficient solvers and scaling TimeFlow to large-scale applications.

## 7 ABOUT THE USE OF LLM

During the writing process, I made limited use of a large language model (LLM) as a tool for language refinement, such as improving clarity, grammar, and academic style. The ideas, analyses, and conclusions presented in this paper remain solely the result of my independent work.

## 8 ETHICS STATEMENT

We confirm that this work aligns with accepted ethical standards in machine learning research. All data and methodologies used are publicly available or properly cited.

## 9 REPRODUCIBILITY STATEMENT

To support reproducibility, we have provided full details of our experimental setup, including hyper-parameters and dataset descriptions, in the experimental section. Code is available.

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

## A  FLOW MATCHING

In this section, we provide a brief overview of flow matching (FM), a generative modeling framework based on learning deterministic vector fields. On a high level, flow matching learns a transport process, where the traditional formulation is based on an ordinary differential equation (ODE) that maps a simple base distribution to a complex data distribution.

Formally, let the target data distribution be $p_{\text{data}}(x)$ on $\mathbb{R}^d$, and let the base distribution $p_0(x)$ be chosen as a Gaussian. Flow matching introduces a continuous interpolation between $p_0$ and $p_{\text{data}}$, where $t \in [0, 1]$ denotes the time variable. A simple choice of interpolation is the linear path.

$$x_t = (1 - \alpha_t)x_0 + \alpha_t x_1, \quad \alpha_t \in [0, 1], \tag{9}$$

where $x_0 \sim p_0$, $x_1 \sim p_{\text{data}}$, and $\alpha_t$ is a schedule function (e.g. $\alpha_t = t$).

The corresponding velocity field of this path is given by

$$v(x_t, t) = \frac{d}{dt}x_t = \dot{\alpha}_t(x_1 - x_0). \tag{10}$$

Flow matching trains a neural network $v_\theta(x_t, t)$ to approximate the true velocity field. The learning objective is to minimize the squared error between the model prediction and the ground-truth velocity:

$$\mathcal{L}_{\text{CFM}}(\theta) = \mathbb{E}_{x_0 \sim p_0, \, x_1 \sim p_{\text{data}}, \, t \sim \mathcal{U}[0,1]} \Big[ \|v_\theta(x_t, t) - \dot{\alpha}_t(x_1 - x_0)\|^2 \Big]. \tag{11}$$

In the ODE formulation, the generative process can be represented in an integral form. Starting from an initial sample $x_0 \sim p_0$, the final output $x_1$ is obtained by accumulating the velocity field along the trajectory:

$$x_1 = x_0 + \int_0^1 v_\theta(x_t, t)\, dt, \quad x_0 \sim p_0, \tag{12}$$

which yields samples approximately distributed according to $p_{\text{data}}$.

Beyond the ODE formulation, flow matching can also be extended to a stochastic differential equation (SDE) by introducing a diffusion term into the dynamics. The forward dynamics are defined as:

$$dx_t = v_\theta(x_t, t)\, dt + \sigma\, dW_t, \tag{13}$$

where $dW_t$ denotes a standard Wiener process and $\sigma > 0$ controls the diffusion strength. Compared to the deterministic ODE case, this stochastic variant allows the model to capture uncertainty and generate diverse trajectories.

The generative process then consists of solving the learned SDE from $t = 0$ to $t = 1$, starting from $x_0 \sim p_0$:

$$x_1 = x_0 + \int_0^1 \big( v_\theta(x_t, t)\, dt + \sigma\, dW_t \big). \tag{14}$$

## B  EXPERIMENTS DETAILS

### B.1  BASELINES

For unconditional generation, we compare TimeFlow with six representative models. Diffusion-TS (Yuan & Qiao, 2024) introduces an interpretable design that disentangles temporal semantics. TimeGAN (Yoon et al., 2019) integrates adversarial training with a supervised embedding to preserve temporal dynamics. TimeVAE (Desai et al., 2021) proposes an interpretable structure tailored to time series data. DiffWave (Kong et al., 2020), originally developed for sequential audio synthesis, has been adopted for general time series modeling. DiffTime (Coletta et al., 2023) decouples constraint specification from training, enabling flexible inference-time adaptation. Cot-GAN (Xu et al., 2020) improves stability through causal optimal transport and entropic regularization. For conditional generation, we benchmark against Diffusion-TS (Yuan & Qiao, 2024) and CSDI (Tashiro et al., 2021), the latter extending diffusion models to conditional tasks via self-supervised masking.

## B.2 EVALUATION METRICS

**Discriminative & Predictive score.** The discriminative score is calculated as $|\text{accuracy} - 0.5|$, while the predictive score is the mean absolute error (MAE) evaluated between the predicted values and the ground-truth values in test data. For a fair comparison, we reuse the experimental settings of TimeGAN (Yoon et al., 2019) for the discriminative and predictive score. Both the classifier and the sequence-prediction model use a 2-layer GRU-based neural network architecture.

**Context-FID score.** A lower FID score means the synthetic sequences are distributed closer to the original data. Paul et al. (2022) proposed a Fréchet inception distance (FID)-like score, Context-FID (Context–Fréchet Inception distance), by replacing the Inception model of the original FID with a time-series representation learning method called TS2Vec (Yue et al., 2022). They have shown that the lowest scoring models correspond to the best-performing models in downstream tasks and that the Context-FID score correlates with the downstream forecasting performance of the generative model. Specifically, we first sample synthetic time series and real time series, respectively. Then we compute the FID score of the representations after encoding them with a pre-trained TS2Vec model.

**Correlational score.** Following Ni et al. (2020), we estimate the covariance of the $i^{\text{th}}$ and $j^{\text{th}}$ feature of a time series as

$$\text{cov}_{i,j} = \frac{1}{T}\sum_{t=1}^{T} x_t^{(i)} x_t^{(j)} - \left(\frac{1}{T}\sum_{t=1}^{T} x_t^{(i)}\right)\left(\frac{1}{T}\sum_{t=1}^{T} x_t^{(j)}\right). \tag{22}$$

Then the metric on the correlation between the real data and synthetic data is computed by

$$\frac{1}{10}\sum_{i,j}^{d} \left| \frac{\text{cov}_{i,j}^{r}}{\sqrt{\text{cov}_{i,i}^{r}\,\text{cov}_{j,j}^{r}}} - \frac{\text{cov}_{i,j}^{f}}{\sqrt{\text{cov}_{i,i}^{f}\,\text{cov}_{j,j}^{f}}} \right|, \tag{23}$$

where superscripts $r$ and $f$ denote statistics computed on real and synthetic data, respectively.

## B.3 DATASETS

table 4 shows the statistics of the datasets and all datasets are available online via the link.

Table 4: Dataset Details

| Dataset | # of Samples | dim | Link |
|---------|--------------|-----|------|
| Sines | 10000 | 5 | https://github.com/jsyoon0823/TimeGAN |
| Stocks | 3773 | 6 | https://finance.yahoo.com/quote/GOOG |
| ETTh | 17420 | 7 | https://github.com/zhouhaoyi/ETDataset |
| MuJoCo | 10000 | 14 | https://github.com/Hdeepmind/dm.control |
| Energy | 19711 | 28 | https://archive.ics.uci.edu/ml/datasets |
| fMRI | 10000 | 50 | https://www.fmrib.ox.ac.uk/datasets |

## B.4 HYPERPARAMETER TUNING AND SENSITIVITY

We conducted all experiments on a single NVIDIA RTX 4090 GPU. The detailed hyperparameters for each dataset are summarized in Table 5. Across datasets, the number of attention heads is fixed to 4, while the head dimension ranges from 16 to 24 depending on the task. The encoder layers vary between 1 and 5, with a batch size of either 64 or 128. We set the number of sampling steps to 100, and the total training steps range from 10,000 to 25,000. In all of our experiments, we use *cosine* noise scheduling and optimize our network using Adam with $(\beta_1, \beta_2) = (0.9, 0.96)$. A linearly decayed learning rate starts at 0.0008 after 500 iterations of warmup. For conditional generation, we set the inference steps 100. We use 90% of the dataset for training and the remaining for testing.

Table 5: Hyperparameters, training details, and compute resources used for each model

| Parameter | Sines | Stocks | ETTh | MuJoCo | Energy | fMRI |
|---|---|---|---|---|---|---|
| attention heads | 4 | 4 | 4 | 4 | 4 | 4 |
| attention head dimension | 16 | 16 | 16 | 16 | 24 | 24 |
| encoder layers | 5 | 2 | 5 | 3 | 4 | 1 |
| batch size | 128 | 64 | 128 | 128 | 64 | 64 |
| sampling steps | 100 | 100 | 100 | 100 | 100 | 100 |
| training steps | 12000 | 10000 | 18000 | 14000 | 25000 | 15000 |

## C  ALGORITHM

Algorithms 1 and 2 present the sampling schemes for conditional and unconditional generation in TimeFlow. For conditional generation, the target sequence is partially observed and a binary mask enforces the known entries at each step. The unobserved parts are initialized with Gaussian noise and then iteratively refined through interpolation between noise and observations, guided by the velocity field predicted by the flow model. This procedure ensures consistency with available information while allowing flexibility to capture stochastic variations. For unconditional generation, the process starts purely from Gaussian noise. The sequence is evolved iteratively using an Euler-type scheme, where the velocity field provides deterministic drift and an additional stochastic term introduces variability. This allows the model to generate realistic sequences that reflect intrinsic temporal uncertainty, even in the absence of conditioning information.

---

**Algorithm 1: Conditional generation**

**Require:** Target time series $\mathbf{Z}_1$, observation mask $\mathbf{M}$, sampling iterations $N$, adaptive parameter $k$, trained flow m model $G_\theta$

1: $\hat{\mathbf{Z}}_1 \sim \mathcal{N}(0, I)$
2: **for** $i = 0$ **to** $N - 1$ **do**
3:    $t_i = (i/N)^k$
4:    $Z_0 \sim \mathcal{N}(0, I)$
5:    $\hat{\mathbf{Z}}_1 = \hat{\mathbf{Z}}_1 \odot (1 - \mathbf{M}) + \mathbf{Z}_1 \odot \mathbf{M}$
6:    $\hat{\mathbf{Z}}_{t_i} = t_i \hat{\mathbf{Z}}_1 + (1 - t_i) Z_0$
7:    $\hat{v}_{t_i} = G_\theta(\hat{\mathbf{Z}}_{t_i}, t_i)$
8:    $\hat{\mathbf{Z}}_1 = \hat{\mathbf{Z}}_{t_i} + (1 - t_i) \hat{v}_{t_i}$
9: **end for**
10: **return** $\hat{\mathbf{Z}}_1$

---

**Algorithm 2: Unconditional generation**

**Require:** Sampling iterations $N$, adaptive parameter $k$, trained flow matching model $G_\theta$
**Ensure:** $\hat{Z}_1$

1: $\hat{Z}_0 \sim \mathcal{N}(0, I)$
2: $t_0 = 0$
3: **for** $i = 0$ **to** $N - 1$ **do**
4:    $t_{i+1} = ((i + 1)/N)^k$
5:    $\hat{v}_{t_i} = G_\theta(\hat{Z}_{t_i}, t_i)$
6:    $\hat{Z}_{t_{i+1}} = \hat{Z}_{t_i} + (t_{i+1} - t_i)\hat{v}_{t_i} + \sqrt{t_{i+1} - t_i}\,\epsilon, \quad \epsilon \sim \mathcal{N}(0, I)$
7: **end for**
8: **return** $\hat{Z}_1$

---

## D  VISUALIZATION

We provide additional illustrative examples of conditional and unconditional generation to further demonstrate the effectiveness of TimeFlow.

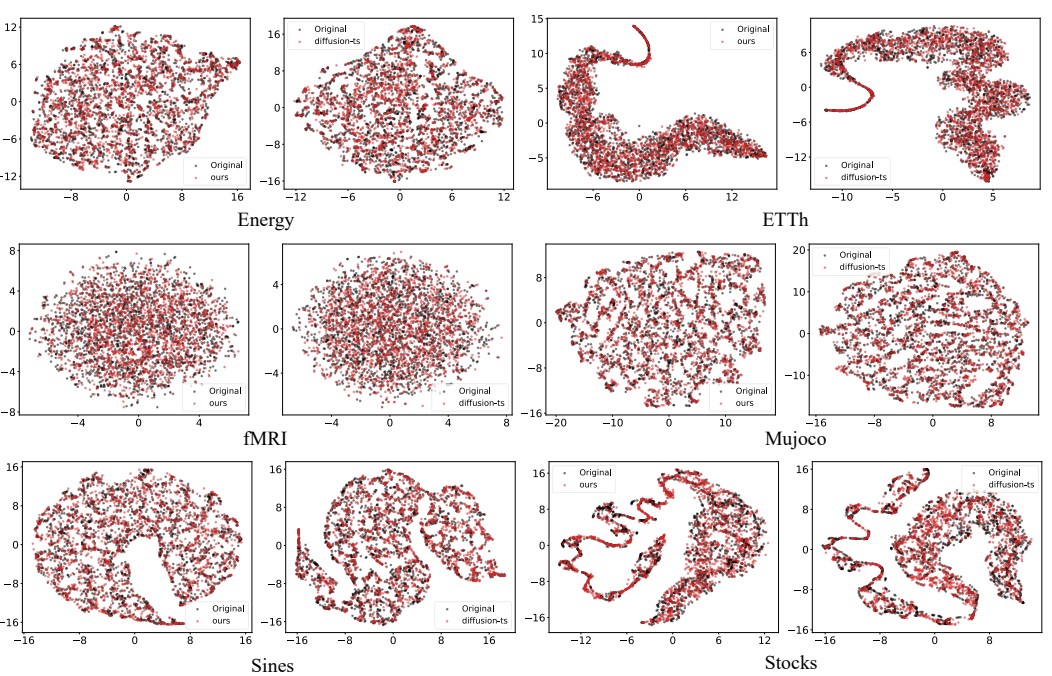

Figure 9: More t-SNE visualizations of the time series synthesized by TimeFlow and Diffusion-TS.

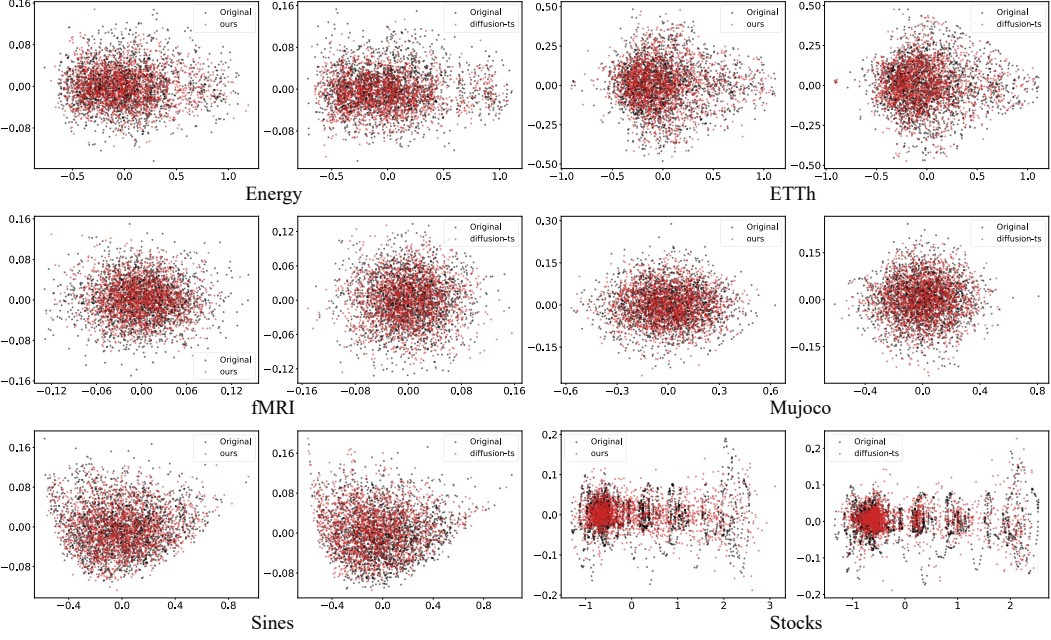

Figure 10: More PCA visualizations of the time series synthesized by TImeFlow and Diffusion-TS.

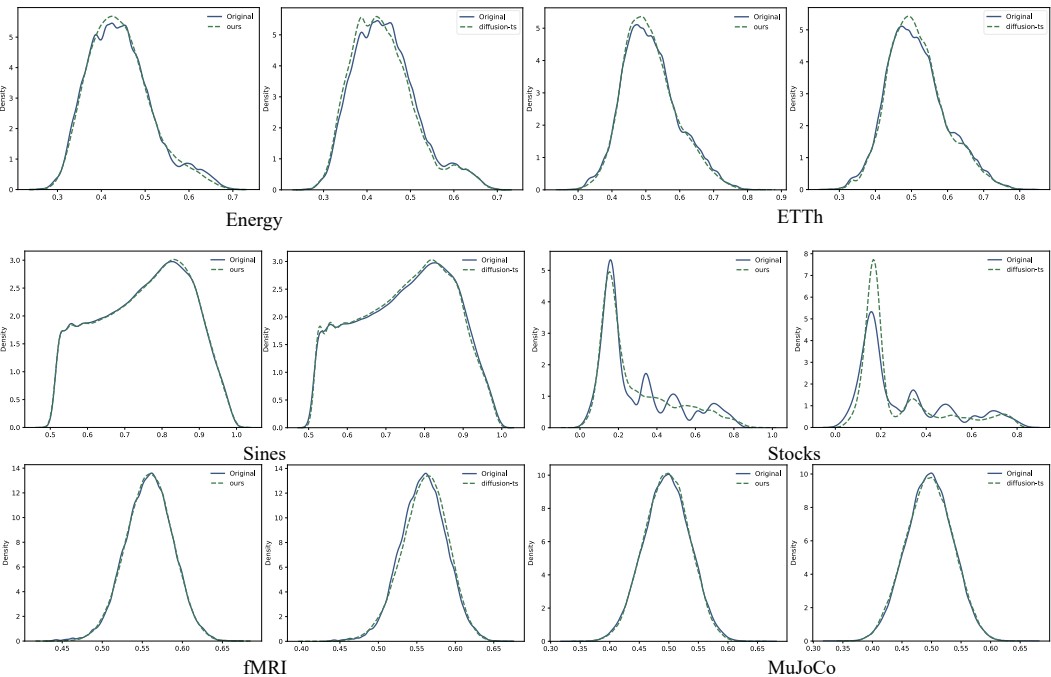

Figure 11: More kernel density estimation visualizations of the time series synthesized by TimeFlow and Diffusion-TS.

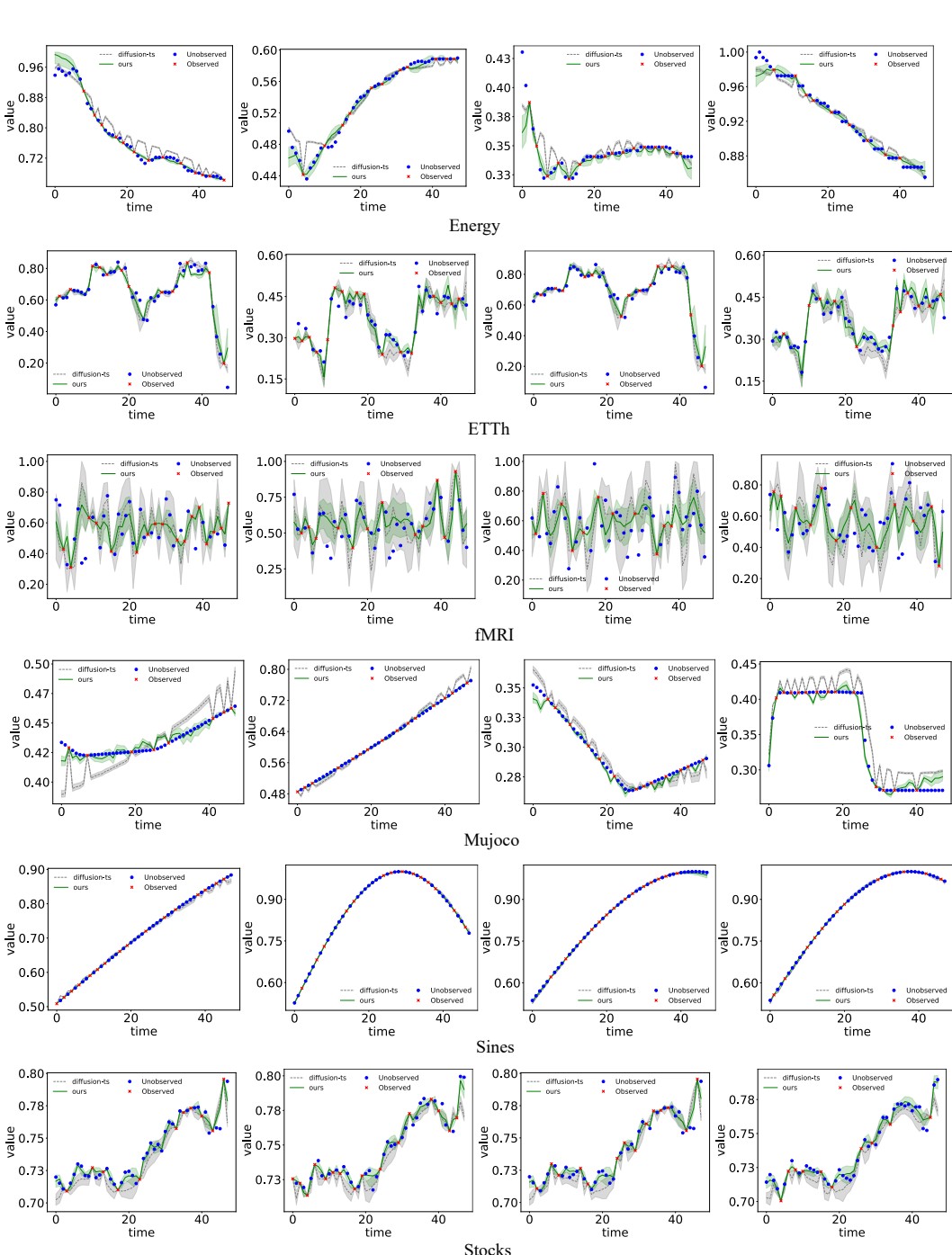

Figure 12: Imputation visualizations of the time series synthesized by TimeFlow and Diffusion-TS on six datasets.

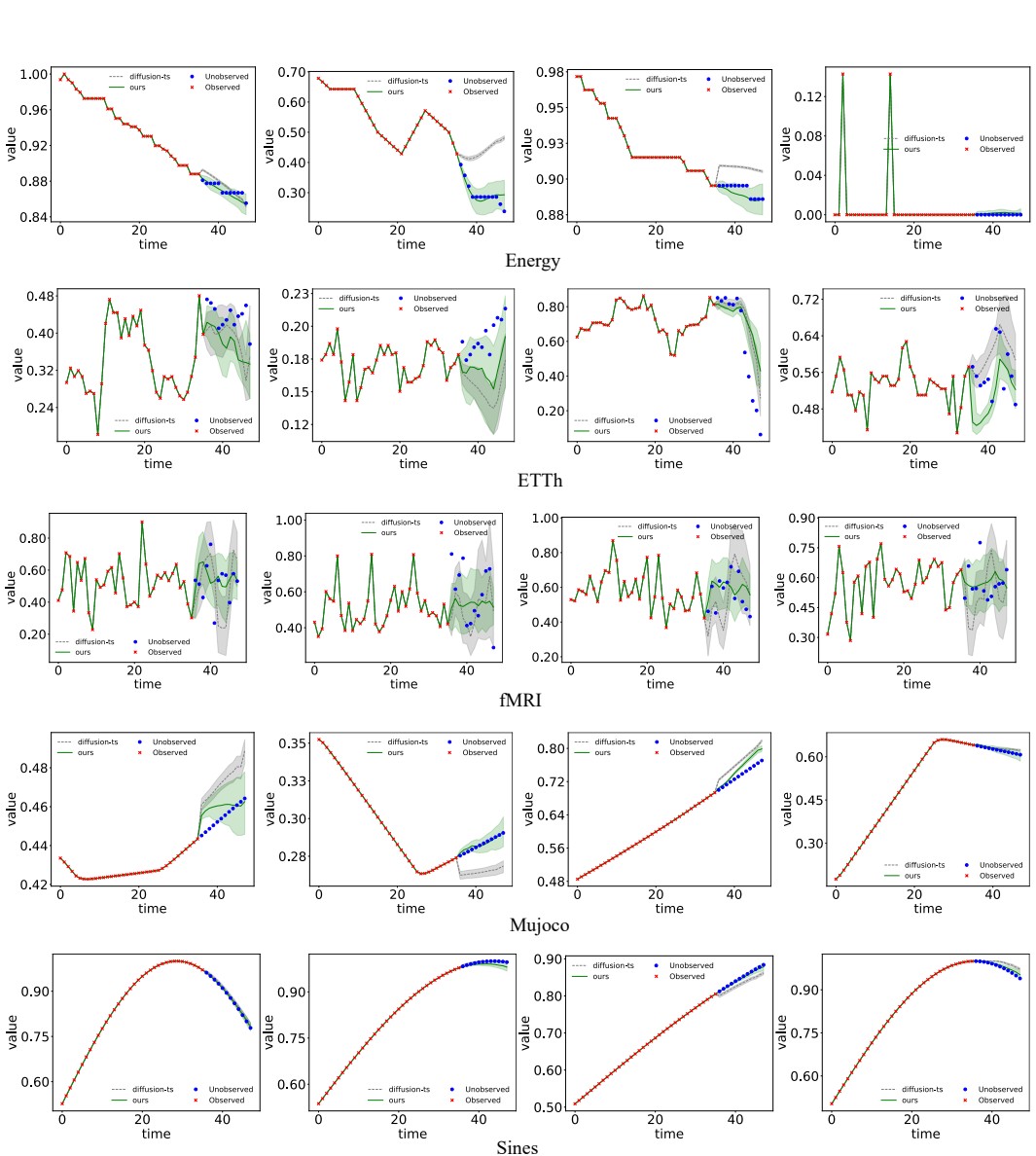

Figure 13: Forecating visualizations of the time series synthesized by TimeFlow and Diffusion-TS on five datasets.

