# OpenReview forum: "TimeFlow: Towards Stochastic-Aware and Efficient Time Series Generation via Flow Matching Modeling"
_ICLR.cc/2026/Conference — ICLR 2026 Conference Withdrawn Submission_

### Official Review · Reviewer_xsSZ · 2025-10-27

**Soundness:** 3
**Presentation:** 3
**Contribution:** 3
**Rating:** 4
**Confidence:** 4

**Summary:**

This paper proposes TimeFlow, an SDE-based FM framework for unconditional time series generation. Compared to standard DDPM-based diffusion models, FM is more efficient with similar performance.

**Strengths:**

1. This paper introduces TimeFlow, an SDE-based FM framework for unconditional time series generation.
2. The paper is well written and easy to follow.
3. Comprehensive experiments are conducted to better understand the performance compared to previous SOTA baselines.

**Weaknesses:**

1. The GitHub link to code doesn’t contain the most important files (models, losses, encoder).
2. Based on the ablation study, CA and FD don’t seem to alter the performance so much. Not sure if those modules are necessary.
3. The discussion of FM seems to be lightweight; the story could be more compelling with a detailed comparison between FM and the diffusion model.
4. Minor: In Figure 3a, both pictures have diffusion-ts as legend.

**Questions:**

See weaknesses

---

### Official Review · Reviewer_TaLf · 2025-10-29

**Soundness:** 2
**Presentation:** 3
**Contribution:** 2
**Rating:** 4
**Confidence:** 4

**Summary:**

The author proposes a novel flow-matching framework TimeFlow with stochasticity concept included for time series generation task, which achieves the competitive performance of efficiency and effectiveness comparisons.

**Strengths:**

1. The motivation is clear and the challenges are almost addressed.

2. The paper is easy to follow  and less typos.

3. The experimental results show  the competitive generation performance and efficiency comparison across the baselines.

4. The process of framework is clear.

**Weaknesses:**

1. In the line 266 “Conditional Generation”, there is no observation values as condition in the TimeFlow, how to achieve the training and generation process (Eqn. 8). Can you give the detailed description of Conditional Generation.

2.According to the Figure 5, with the forecasting window longer, the mse is much similar to the baseline,  and the growth rate far exceeds the corresponding baseline, which demonstrates the instability of TimeFlow in forecasting task. Can you give some clarification.

3. According to section 5.4, there is no further conclusion for the role of diffusion coefficient. Can you show some deeper conclusion？

4. Some strong baselines are not included in the paper, such as KoVAE [1], PaD-TS [2]. Please add to verify the effectiveness.

[1]. Naiman, Ilan, et al. "Generative Modeling of Regular and Irregular Time Series Data via Koopman VAEs." The Twelfth International Conference on Learning Representations.

[2]. Li, Yang, et al. "Population Aware Diffusion for Time Series Generation." Proceedings of the AAAI Conference on Artificial Intelligence. Vol. 39. No. 17. 2025.

**Questions:**

Please refer to the weaknesses.

---

### Official Review · Reviewer_KCM6 · 2025-10-29

**Soundness:** 1
**Presentation:** 2
**Contribution:** 1
**Rating:** 0
**Confidence:** 4

**Summary:**

TimeFlow is a "SDE flow-based" generative frame work for time series, which aims at bridging the gap between Diffusion based and Flow Matching based models in terms of efficiency, expressivity and ability to capture the stochasticity of time series.

**Strengths:**

- The paper is well written in structure.
- The proposed parametrization of the model is reasonable.

**Weaknesses:**

In short, the paper lacks novelty and to me seems to misrepresent its contribution.

Wrong motivation and falsely claimed contribution:
- They argue that Diffusion models (including SDE base score models, where the score matching objective actually motivated the flow matching framework) are inefficient, when compared to the ODE of flow models. Then acknowledge that the ode lacks the capacity to model the stochasticity of time series, leading to their conclusion to turn the ODE into an SDE and still call it flow matching. However, the proposed SDE is just another SDE-based diffusion model.

Experiments:
- Do not compare to state of the art flow and diffusion models (e.g., [1,2,3]).
- Efficiency claims are not substanciated properly. Compare you SDE flow against diffusion SDE with same architecture. Further, how does your SDE (the reason why Diffusion is more inefficient than flow matching) compare against a flow ode. Can you sample you method as an ODE?



1. Modeling temporal data as continuous functions with stochastic process diffusion, ICLR 2023
2. Predict, refine, synthesize: Self-guiding diffusion models for probabilistic time series forecasting, NeurIPS 2023
3. Flow Matching with Gaussian Process Priors for Probabilistic Time Series Forecasting, ICLR 2025

**Questions:**

- Can you explain the difference of your SDE-based flow model to a score based model? Be specific and formal, how does the SDE-based flow connect to score-based SDEs (e.g., Song et al., 2021).
- If your model is a flow model, by definition it should be invertible. How can you invert your method? How could a SDE be invertible?
- How does the SDE modify the underlying continuity equation?

---

### Official Review · Reviewer_8zFa · 2025-11-02

**Soundness:** 3
**Presentation:** 3
**Contribution:** 1
**Rating:** 0
**Confidence:** 5

**Summary:**

The paper proposed TimeFlow, a novel SDE-based flow matching framework for time series generation.

**Strengths:**

This paper designed a component-wise decomposed velocity field to capture the multi- faceted structure of time series and seem to contribute to the performance improvement.

**Weaknesses:**

This idea to use flow matching has been adopted in FlowTS https://arxiv.org/pdf/2411.07506. The idea is presented much simpler and effective in this paper. So the novelty of this paper is limited.
Most importantly, the authors failed to do enough literature review to find the paper published one year ago and did not include any comparisons with this papers.
After careful comparing the table 1 of both papers, I think their performances of these two papers are quite similar. Therefore, I think there is no need to restudy the time series via flow matching again in this paper.
In addition, this paper also fail to study the unconditional long-term time series generation.
Lastly, the open source efforts by this paper is limited, since the provided link did not provide any README for how to train/eval this code.
Therefore, I would give strong rejection for this paper.

**Questions:**

What is the important contribution of this paper compared to FlowTS?
Also, why did this paper not include FlowTS into comparison?

---

### Note · Authors · 2025-11-18

I have read and agree with the venue's withdrawal policy on behalf of myself and my co-authors.